# The Role of Iron and Zinc in the Treatment of ADHD among Children and Adolescents: A Systematic Review of Randomized Clinical Trials

**DOI:** 10.3390/nu13114059

**Published:** 2021-11-13

**Authors:** Roser Granero, Alfred Pardo-Garrido, Ivonne Lorena Carpio-Toro, Andrés Alexis Ramírez-Coronel, Pedro Carlos Martínez-Suárez, Geovanny Genaro Reivan-Ortiz

**Affiliations:** 1Department of Psychobiology and Methodology, Autonomous University of Barcelona, 08193 Barcelona, Spain; Alfredo.Pardo@uab.cat; 2Ciber Fisiopatología Obesidad y Nutrición (CIBERobn), Instituto Salud Carlos III, 28029 Madrid, Spain; 3Basic Psychology, Behavioral Analysis and Programmatic Development PAD-Group, Catholic University of Cuenca, Cuenca 010107, Ecuador; greivano@ucacue.edu.ec; 4Laboratory of Basic Psychology, Behavioral Analysis and Programmatic Development PAD-Lab, Catholic University of Cuenca, Cuenca 010107, Ecuador; icarpiot@ucacue.edu.ec (I.L.C.-T.); andres.ramirez@ucacue.edu.ec (A.A.R.-C.); pmartinezs@ucacue.edu.ec (P.C.M.-S.); 5Laboratory of Psychometry, Comparative Psychology and Ethology, Catholic University of Cuenca, Cuenca 010107, Ecuador; 6Health and Behavior Research Group (HBR), Catholic University of Cuenca, Cuenca 010107, Ecuador

**Keywords:** ADHD, zinc, iron, treatment, children

## Abstract

Attention-deficit/hyperactivity disorder (ADHD) is a neurodevelopmental disorder common from childhood to adulthood, affecting 5% to 12% among the general population in developed countries. Potential etiological factors have been identified, including genetic causes, environmental elements and epigenetic components. Nutrition is currently considered an influencing factor, and several studies have explored the contribution of restriction and dietary supplements in ADHD treatments. Iron is an essential cofactor required for a number of functions, such as transport of oxygen, immune function, cellular respiration, neurotransmitter metabolism (dopamine production), and DNA synthesis. Zinc is also an essential trace element, required for cellular functions related to the metabolism of neurotransmitters, melatonin, and prostaglandins. Epidemiological studies have found that iron and zinc deficiencies are common nutritional deficits worldwide, with important roles on neurologic functions (poor memory, inattentiveness, and impulsiveness), finicky appetite, and mood changes (sadness and irritability). Altered levels of iron and zinc have been related with the aggravation and progression of ADHD. Objective: This is a systematic review focused on the contribution of iron and zinc in the progression of ADHD among children and adolescents, and how therapies including these elements are tolerated along with its effectiveness (according to PRISMA guidelines). Method: The scientific literature was screened for randomized controlled trials published between January 2000 to July 2021. The databases consulted were Medline, PsycINFO, Web of Science, and Google Scholar. Two independent reviewers screened studies, extracted data, and assessed quality and risk of bias (CONSORT, NICE, and Cochrane checklists used). Conclusion: Nine studies met the eligibility criteria and were selected. Evidence was obtained regarding the contribution of iron-zinc supplementation in the treatment of ADHD among young individuals. The discussion was focused on how the deficits of these elements contribute to affectation on multiple ADHD correlates, and potential mechanisms explaining the mediational pathways. Evidence also suggested that treating ADHD with diet interventions might be particularly useful for specific subgroups of children and adolescents, but further investigations of the effects of these diet interventions are needed.

## 1. Introduction

Attention-deficit hyperactivity disorder (ADHD) is a neurodevelopmental disorder, usually beginning in early childhood and with a chronic progression to adulthood with several negative consequences, such as low self-esteem, difficulties in interpersonal relationships, and problems in school learning [1]. Subtypes according to the Diagnostic and Statistical Manual of Mental Disorders (5th edition) are inattention (being unable to keep focus), hyperactivity (excess movement that is not fitting to the setting) and impulsivity (hasty acts that occur without thought) [2]. Epidemiological studies indicate that ADHD is one of the most common mental disorders affecting children [3,4], with prevalence in the childhood–adolescent community ranging between 2% and 8% globally [5,6]. High prevalence rates persist in adult ADHD patients, into the range of 2.6% to 6.8% [7,8].

Etiological research outlines that ADHD constitutes a complex condition with multiple interactive factors [9]. Specific causes of the disorder are yet to be determined, but various contributing risk factors have been identified, including genetics [10,11,12,13], epigenetics [14,15], problems during pregnancy (such as stress, substance use, and other mental and physical diseases) [16,17,18], premature birth [19,20], obstetric and neonatal complications [21,22,23,24], and infections [25,26]. Neuropsychological mechanisms have also been identified in the onset and course of ADHD, such as brain injuries [27], neuroanatomical substrates related with genuine motor dysfunction [28] and deficient decision-making [29]. Diet-related factors (such as vitamin and/or mineral deficiencies) have also been evidenced to contribute to the levels of ADHD and the progression of the disease [30,31,32]. 

Diverse treatment plans have demonstrated to improve the symptoms of ADHD, usually combining medication (stimulants and non-stimulants have proved efficacy) with behavioral therapy [33,34]. Empirical-based pharmacological plans that have been used for decades include methylphenidate, amphetamine, atomoxetine, and guanfacine [35,36]. However, some children using these medications only experience partial relief of symptoms, which persist even with a change in medication or adjustment in dose [37]. Additionally, a number of patients with medication report significant adverse effects, such as loss of appetite, irritability, and sleep disorders [38,39]. Other typical ADHD correlates, such as restless leg syndrome, also persist despite medication [40,41].

On the other hand, despite growing pharmacological discoveries, mental disorders (such as ADHD) have shown a gradual rise during the last few decades, and rates are expected to continue increasing in the coming years [42,43,44]. Interestingly, it has been observed that worsening mental health within developed countries could be related to diet habits (more precisely, to the transition to more calorically dense ones) and to lower levels of physical activity [45]. Centered in the area of ADHD, various studies have shown harmful effects of the diet in the onset and progression of ADHD, including the use of preservatives and food additives [46,47]. Within this research area, it has also been observed that changes in the diet style could improve treatment efficiency for ADHD, particularly the utilization of some minerals and vitamins [48]. 

Dietary models have proved to be relevant to both metabolic performance and individuals’ behavior. Restriction and elimination diets have been tested in ADHD treatments [49], and supplements and nutritional products have also been used as complementary intervention plans [50,51]. Studies assessing the specific role of iron and zinc on the course of ADHD have observed the deficiency of both minerals in most cases of hyperactivity [52]. Zinc is an essential mineral involved in numerous cellular metabolic processes, required for the catalytic activity of a large number of enzymes, implied in the accurate functioning of the immune system and with a role in the protein synthesis, DNA synthesis, and cell division [53]. Zinc also contributes to normal growth and development from pregnancy to adolescence, and harmonizes the performance of dopamine and melatonin [54]. Iron is a cofactor mineral that systematizes the production of dopamine and norepinephrine, an essential element participating in a wide variety of metabolic processes, including oxygen transport, deoxyribonucleic acid synthesis, and electron transport [55]. 

Some studies have analyzed the contribution of zinc and iron on ADHD occurrence, but conflicting findings have been obtained [56]. As a trend, compared to healthy control groups, lower levels of iron and zinc have been observed in children diagnosed with ADHD, but it is not clear whether changes in nutrient levels in blood tests mediate treatment outcomes in children with ADHD who consume mineral supplements [57,58], or even what sub-groups could particularly benefit from. 

On the basis of the controversial results published regarding the role of dietary nutrients with zinc and iron for the treatment of ADHD [59,60], we performed this qualitative systematic review aimed to provide an updated account of the evidence published in randomized clinical trials assessing the efficacy of both supplements in the treatment of ADHD among children and adolescents.

## 2. Materials and Methods

### 2.1. Procedure

This systematic review was done in accordance with the eligible criteria reported in the *Preferred Reporting Items for Systematic Reviews and Meta-Analysis* (PRISMA) [61] (see Appendix A, Appendix A), the *Assessment of Multiple Systematic Review* (AMSTAR) [62] and the *Cochrane Handbook for Systematic Reviews of Interventions* [63].

### 2.2. Inclusion and Exclusion Criteria

Medline, PsychINFO, Web of Science, and Google Scholar databases were searched. The search was further limited to the next inclusion criteria: (1) studies which examined treatments for ADHD; (2) which were based on randomized-controlled clinical trials; (3) which were published in English/Spanish language; (4) which were published in peer-reviewed Journals between January 2000 and July 2021; (5) which aimed to assess the efficacy of a nutritional intervention through iron or zinc supplementation; (6) which assessed clinical samples of ADHD patients diagnosed according to the Diagnostic and Statistical Manual of Mental Disorders (DSM) criteria, the 4th [64] or 5th [2] versions; (7) which analyzed participants with an age range between 5 to 18 years old; and (8) which used validated self-report instruments for measuring ADHD-related problems before and after treatment. 

Specifically excluded from this systematic review were non-pill-based treatment modalities, such as behavioral interventions, neurofeedback, restriction, or alternative food exclusion diets or chiropractic interventions. No restrictions were considered for the longitudinal follow-up and interventions for the control group. 

### 2.3. Research and Selection of Studies

The research was conducted on 20 July 2021. The strategy research (keywords and search sequence) for each database was:Medline: search = [((zinc OR iron) AND (ADHD)) AND ((treatment or therapy))]. The next filters were employed: (a) type of publication: (Clinical Trial) and (Randomized Controlled Trial); and (b) Date of publication: (From 1 January 2000–31 July 2021).PsycINFO: search = [((zinc OR iron) AND (ADHD)) AND ((treatment or therapy))]. The next filters were defined: (a) Type of publication: [Peer Reviewed Journal]; and (b) Date of publication: [1 January 2000–31 July 2021];Web of Science: search = [(((zinc OR iron OR ferritin) AND (ADHD)) AND ((treatment or therapy))))]. The next filters were defined: (a) Document Types: [Clinical Trial], and (b) Publication years: [1 January 2000–31 July 2021];Google Scholar: search = [(zinc OR iron) AND (supplement) AND (ADHD) AND (treatment or therapy) AND (randomized or “clinical trial”) AND (children or adolescent) AND (“Peer Reviewed Journal”)].

Two authors of this systematic review independently analyzed the title and abstract of each record, according to the inclusion/exclusion criteria. Only studies meeting the eligibility criteria were then extracted. Data validation was discussed by the same authors, and disagreements were resolved by discussion and if needed by consulting a third team member until a consensus was reached.

The authors also extracted data of the identified screening records to be described and evaluated in the results section. The information considered for applying the eligibility criteria was: the date and location of publication, type of publication, sample size, participants’ sex and age range, study design, type and duration of the treatment, and measures and outcomes (changes in the ADHD status, metabolic levels, and other psychological areas).

### 2.4. Study Quality Assessment

The *Consolidated Standards of Reporting Trials* guidelines were employed (CONSORT-2010) [65]) for assessing the study quality. This checklist is used worldwide to improve reported randomized controlled clinical trials through a list of 25 items for assessing the title (inclusion of the design type), elaboration of the abstract (structured and completed), background and explanation of the rationale, definition of the objectives and hypothesis, description of the trial design (including important changes to methods after trial commencement and reasons), eligibility criteria for participants, the setting and location where the data were collected, intervention description (sufficient details to allow replication), completely defined outcome measures, sample size calculation (or power analysis), the method used to generate the random allocation sequence (including type of randomization), use of blinding methods, statistical procedures used for the analyses, results description (including comparison at baseline), discussion of the results (including limitations and generalizability), and other information (registration, protocol, and funding).

In addition, two checklists were used. First, the guidance elaborated by the National Institute for Health and Care Excellence: The NICE Methodology Checklist for Randomized Controlled Trials [66], which includes items structured in four sections: (a) assessment of the selection bias (appropriate method of randomization used, adequate concealment of allocation and comparability at baseline); (b) assessment of the performance bias (the comparison groups received the same care apart from the interventions, and using double-blind method); (c) assessment of the attrition bias (the groups followed for an equal length of time, and comparable treatment completion); and (d) assessment of detection bias (adequate length of follow-up, use of precise definition for the outcomes, valid and reliable methods for measuring the outcomes and triple-blind method used). Second, the Critical Appraisal Skills Programme (CASP) Randomized Controlled Trial Standard Checklist [67], which includes a set of items organized in four sections: (a) validity of the study design (clearly focused research question, randomized assignment of the participants to groups and complete follow-up); (b) using adequate methodology procedures (blinded methods, similarity between the groups at the start of the controlled trial, and groups receiving the same level of care); (c) results reported comprehensively, including the estimate of the effect sizes and carrying out a cost-effectiveness analysis cost; and 4) applicability of the results (generalization features and value provided by the intervention).

The risk of bias was also assessed using the Cochrane Risk of Bias 2 checklist [68], based on seven domains: (a) the randomization process, (b) deviations from intended intervention/s, (c) missing outcome data, (d) outcome measurement/s, (e) selection of the reported results, (f) incomplete reporting, and (g) power calculation (or sample size justification).

The assessment of the methodological quality was rated by authors of this systematic review, and discrepancies were discussed and solved.

## 3. Results

### 3.1. Descriptive for the Selected Studies

The number of studies identified through database-searching was *n* = 123. After removing duplicate articles, *n* = 97 studies were screened and *n* = 9 were finally selected on the basis of the title-abstract and the eligibility criteria (Figure 1 contains the search flow-chart).

The randomized clinical trials included in the synthesis reported data for ADHD samples of children and adolescents aged 5 to 15 years old. Zinc sulfate was administered in *n* = 5 studies, iron in *n* = 2 studies, and multi-supplements containing both compounds in a *n* = 2 study. Clinical improvement was verified using standardized measurement tools for versions for parents, teachers, and/or children. Laboratory blood screening tests were also applied (at least) at baseline and at the end of the treatment.

Table 1 includes the description of the *n* = 9 randomized controlled trials of supplements with zinc and iron for the treatment of ADHD, such as: identification of the study, sample size (N), supplement administered to the experimental group, age range and mean, gender distribution, duration of the study, and the standardized measures employed for measuring the ADHD symptoms, and other related problems.

### 3.2. Assessment of the Methodological Quality, Adherence and Competence

Table 2, Table 3 and Table 4 contains the results of the assessment with the CONSORT, NICE, and CASP checklists for the *n* = 9 selected studies included in the systematic review. Many studies met most of the criteria. The most incomplete items were the calculation of the sample size (or the power estimation based on the sample size of the groups), carrying out a triple-blinded study [the selected studies were double-blinded (patients and researchers were unaware of whether the treatment was administered), but it was not reported whether the team analyzing the data was also unaware of which groups’ data they were evaluating], obtaining the precision of the estimates (through confidence intervals or other alternative standardized effect size measures) and carrying out a cost-effectiveness analysis cost). 

### 3.3. Assessment of the Risk of Bias of the Included Studies

Table 5 shows the risk of bias summary of the included studies. The signaling question “bias arising from the randomization process” was identified with a moderate risk of bias for all the studies because the allocation sequence was not clearly concealed in the selected studies. The signaling item “statistical power calculation” was judged as low because studies did not justify the sample size before the results section.

### 3.4. Efficacy of Zinc for ADHD Treatment

The study of Noorazar and colleagues used the next intervention [69]: (a) a dose of 0.5–1 mg/kg/day methylphenidate plus placebo in the control group; and (b) a dose of 0.5–1 mg/kg/day methylphenidate plus 10 mg zinc (10 cc zinc sulfate syrup). No differences between the groups were observed regarding the dose of methylphenidate. After 6 weeks of treatment, the authors found that the use of zinc was useful to decrease the inattention scores (*p* = 0.02), but no differences between the groups were found in the other measures for the hyperactivity and impulsivity scales. The lack of differences in the total ADHD score in this study suggested that augmentation with zinc could only partially improve ADHD severity levels. The authors of the study also noticed that the difference between the two groups regarding the gender ratio compromised the generalization capacity, and that the use of the Connors Parent’s Questionnaire as the only measure for ADHD was a significant limitation [although it is a standardized tool, the inclusion of additional instruments and informants (such as teachers) could have provided different perspectives of the children’s behaviors and impairments].

The study of Zamora and colleagues also valued the effect of zinc supplementation (10 mg/day) as an adjuvant therapy (complementary to methylphenidate, with a dose of 0.3 mg/kg/day) for ADHD in *n* = 40 pediatric children, and obtained an improvement in ADHD signs associated to the experimental group in the questionnaires answered by teachers (no differences emerged from the parents’ reports) [70]. These authors also obtained decreased zinc levels during the intervention in both groups (control and experimental conditions), suggesting that the methylphenidate could contribute to a decrease in zinc concentration that can be counteracted with zinc supplementation.

In the research by Arnold et al. with *n* = 52 ADHD children treated with d-amphetamine (weight-standardized) and zinc complementation (15 mg/day or 30 mg/day) or d-amphetamine and a placebo, a similar improvement in ADHD was observed in all the conditions (more precisely, teachers’ ratings showed medium effect sizes favoring zinc, but parents’ reports showed an opposite trend favoring the placebo) [71]. Interestingly, this trial also observed that the group treated with 30 mg/day zinc obtained 37% lower plasma levels of conventional drugs. The authors of this trial concluded that rises in zinc levels in blood tests suggested that children with ADHD may have a zinc-wasting metabolism (defined as low levels of zinc related with potential deficits in absorption or losses in urine), and therefore the zinc administered in the morning acted only as an immediate-release stimulant (this was probably the reason of the teachers’ reports favoring this dietary supplement) but was next immediately excreted by mid-afternoon (parents could not observe improvements on the children’s behaviors). As a global conclusion, Arnold and colleagues indicated that their study did not support zinc supplementation as a complementary tool for ADHD treatments, but also suggested that before discarding zinc as a potential treatment, it would be desirable that future research explores the situation further with different zinc doses/preparations, samples with children characterized for low zinc levels, and measures focused on neuropsychological tests as primary outcomes.

The study carried out by Akhondzadeh et al. aimed to compare two groups of patients with ADHD (with or without zinc augmentation, with a dose of approximately 15 mg/day) and both conditions treated with methylphenidate (1 mg/kg/day), and the experimental group obtained more improvement according to parents’ and teachers’ ratings at 6 weeks of the intervention [72]. Trend analyses also showed that differences between the groups increased during the treatment: while the placebo and the zinc supplementation groups only showed small differences at week 2, moderate to high differences were found at 4 weeks and 6 weeks (this pattern was observed for the questionnaires answered by teachers and parents).

The study by Bilici and colleagues among a large sample of *n* = 400 children with ADHD observed that the group receiving the zinc supplement (approximate dose of 40 mg/day) compared with the control group (receiving placebo) obtained greater improvements in multiple measures (hyperactivity, impulsivity, and socialization symptoms) but results were similar for attention deficiency [73]. The greatest differences between the groups were observed at 12 weeks of the intervention, and were few during 1 to 4 weeks. Regarding the potential mechanisms explaining these results, the authors conclude that the zinc supplement could affect the conversion of dietary pyridoxine to its active form (pyrodoxal phosphate), implied in the process of conversion of tryptophan to serotonin. In this sense, zinc should contribute to the increase in serotonergic functions, contributing to a decrease in the characteristic of impulsivity as a symptom of ADHD. Alternatively, the authors also suggested the existence of a synergism of zinc in regulating dopamine and norepinephrine, which has been implied in ADHD treatments. The study by Bilici et al. also showed that pre/post differences in the ADHD measures were predicted by age and BMI (a positive association was found in the B-slopes in regression analyses as higher age, and the BMI as higher change). Low pre-treatment zinc and free fatty acid values were also associated with a smaller decrease in ADHD symptom levels before and after the treatments. This evidence was also useful to determine which children could benefit more from including zinc supplementations in the ADHD treatments.

### 3.5. Efficacy of Iron for ADHD Treatment

The study by Konofal and colleagues aimed to examine the contribution of iron supplementation (ferrous sulfate, 80 mg/day) in a sample of *n* = 23 children who met the criteria for ADHD, which showed that the experimental group reported more improvement than the control group in different treatment outcomes [74]: ADHD symptom levels reported by patients, parents, and teachers (particularly marked for the inattention factor), and also the presence of restless leg syndrome. The authors did not find a correlation between the baseline levels of serum ferritin levels with endpoint ADHD measures, suggesting that children with the lowest pre-treatment ferritin values did not benefit more from iron supplementation therapy than other children. In their conclusions, these authors suggested that the effectiveness of ferrous sulfate could be associated with the ADHD pathophysiology, that the benefits of restless leg syndrome could be the consequence of the improvement in the ADHD motor activity in the evening, and that iron could enhance the action of pharmacological treatment with methylphenidate and amphetamine.

The study of Panahandeh et al. in a sample of *n* = 42 children with ADHD also obtained benefits for the use of ferrous sulfate (5 mg/kg/day) plus methylphenidate (1 mg/kg/day), in the inattentiveness, hyperactivity, and impulsive symptom levels reported by parents at 2 months of treatment [75]. In the conclusions section, these authors supported the hypothesis that the contribution of iron supplementation on the ADHD improvements could be explained by the capacity of this element in the dopamine transporter density and activity, which is consistent with other studies that observed decreased thalamic iron levels in children with ADHD compared to controls [77,78].

### 3.6. Efficacy of Including Simultaneously Iron and Zinc for ADHD Treatment

Some randomized clinical studies have also valued the contribution of multimineral-vitamin supplements which include zinc and iron for ADHD treatment in children. The rationale for selecting trails with multi-supplements therapy plans for ADHD was the low number of trails using simultaneously zinc and iron, and those identified used preparations with vitamins, dietary minerals and other nutritional elements. Therefore, considering these studies facilitates generalizability of the results related with the use of zinc–iron supplements. 

The study of Rucklidge and colleagues [76] in a sample of *n* = 93 patients used Daily Essential Nutrients (DEN, with Recommended Dietary Allowances, RDA), which contains a comprehensive range of micronutrients (13 vitamins, 17 minerals, and 4 amino acids). The participants were instructed to titrate the dose up to over a week, starting with 3 capsules per day and increasing it to up to 12 capsules per day (see Table 1 for the zinc–iron content). This study obtained an intent-to-treat analysis of between-group differences with greater improvements for ADHD related with the micronutrient treatment, but specifically in the clinicians’ reports assessing inattention. However, although no significant differences were found for the hyperactivity and impulsive measures between the experimental and the placebo groups, the micronutrients supplement obtained improvements in emotional regulation, aggression, and general functioning. The authors of this study concluded that the mechanisms of action of micronutrient treatments likely involves different pathways rather than pharmacological treatments, and that supplements like those used in the study may have an impact on the methylation/methionine cycle, required for the synthesis of DNA/RNA (as it was suggested by other studies [79]), and on the citric acid cycle and electron transport chain acting as co-enzymes in mitochondrial aerobic respiration and energy production. These potential effects on the physical state could also contribute to improvements in mood state and cognition functions (also suggested in the literature [80,81]).

Finally, the study of Rucklidge et al. was published later and carried out on a sample of *n* = 38 children that also used DEN, with an initial dose of 3 capsules per day which further increased to 12 capsules per day (see Table 1 for the zinc–iron content). This study showed that a broad-spectrum micronutrient formula (EMPowerplus) served as a mediator in the treatment response considering multiple ADHD measures [58]. Interestingly, only pre/post changes in the copper and ferritin levels achieved a significant moderator role on the improvement registered for the severity of ADHD: as the increase in the ferritin and the decrease in the copper highly benefited the levels of ADHD (in the three dimensions of inattentiveness, hyperactivity, and impulsivity). However, the authors indicated that these results should be interpreted with caution, and outlined the need to consider individual variability: since metabolic needs are different between the patients, as well as their genetic makeup, some children may need certain nutrients to restore optimal metabolic functioning, and others will not require specific nutrients and should not benefit from supplementations. In addition, nutrients act in a synergistic way, and therefore, changes in one can have cascading effects on others, and it was thus unlikely that increases in a specific nutrient could make a direct and unique contribution on the changes observed in a complex disorder, such as ADHD.

### 3.7. Tolerability Analyses

Regarding safety and tolerability, no serious adverse events were reported in the studies selected for this review, and the unpleasant events found were similar for the experimental and control groups. Iron and zinc therapies were generally well-tolerated: no patients referred to an exacerbation of ADHD symptoms or a significant decrease in appetite, and only a low number of children reported physical pains, nausea, vomiting, constipation, or a metallic taste in the mouth. Moreover, studies suggested that maintaining a fixed daily dose of iron–zinc was a relevant factor in increasing tolerability, and no indication regarding any great concern about the doses up to the follow-up was registered. However, it cannot be determined whether a long-term zinc–iron treatment could induce negative events, such as toxicity or hemosiderosis.

## 4. Discussion

This study conducted a qualitative systematic review of the empirical evidence obtained in randomized clinical trials published since 2000 on the efficacy of zinc and iron supplementation among ADHD children and adolescents. Nine studies met the eligibility criteria and were selected. Results indicated that low zinc and iron levels were related with both higher baseline levels of ADHD severity and poorer treatment outcomes. Compared with the controlled-placebo conditions, the dietary supplements with zinc and iron were associated with improvements in ADHD severity at the end of the treatments, although the effect size of the outcomes tended to be low and/or focused on specific ADHD symptoms/measures.

Cross-sectional evidence regarding the association of zinc and iron with ADHD levels (pre-treatment associations) is consistent with other non-randomized studies that explored the contribution of these micronutrients within clinical- and population-based samples. For example, a longitudinal study with a large sample with *n* = 608 children found that zinc–iron supplementation improved attention/concentration in children over a 14-month follow-up period [82]. A positive correlation was also found between zinc and symptoms of inattentiveness in a cross-sectional study of *n* = 48 children [83]. 

The potential explanations about the paths between the zinc–iron levels and the various ADHD manifestations are discussed in a number of studies, which showed that lower levels of these elements in childhood (but also in older adults) could be related to a dopaminergic impairment, which could be the origin of the high inattentiveness–hyperactivity levels and the other correlates of the ADHD disease (like the presence of restless leg syndrome and other multiple sleep problems) [74,84,85]. Neuroimaging, genetic, and animal studies have also shown that decreases in dopamine transporter densities are related with decreased brain iron and zinc levels [86,87,88]. Zinc achieves a relevant role in the metabolism of neurotransmitters, prostaglandins, and for maintaining brain structure and function. Since zinc is necessary in the metabolism of melatonin, it is strongly implied in the regulation of dopamine (zinc is therefore considered a dopamine reuptake inhibitor). Iron is a cofactor in the tyrosine hydroxylase enzyme, and a key cofactor in the making of neurotransmitters, including serotonin, norepinephrine, and dopamine. Brain iron-deficient rodents have been associated with alterations in adenosine neurotransmission, which has provided a pathogenetic link with glutamate mechanisms and the hypersensitive corticostriatal terminals involved in some ADHD correlates [89,90]. 

The link between iron and zinc supplementation and the observed decrease in ADHD symptoms in the controlled clinical trials selected for this systematic review also makes sense with regard to the role of zinc and iron as dopamine reuptake inhibitors, which was precisely the core target of the stimulant medications used in the combined treatment plans for the disease. Other non-randomized studies using a combination of polyunsaturated fatty acids with zinc supplementation also showed benefits for the treatment of ADHD. For example, the observational study of Huss et al. in a large sample of *n* = 810 monitored children aged 5–12 years old showed that consumption of omega-3 and omega-6 with complementation of zinc improved ADHD symptoms, as well as other emotional problems, and sleeping disease (attention deficit, hyperactivity, and impulsivity) after 12 weeks [91]. The rationale for these results could be found in the capacity of these dietary combinations for facilitating the transmission of signals between neurons, which also contributed to improvements in attention ability, learning difficulties, and executive functions in the development of children and adolescents [92,93].

The studies included in this review support the existence of subgroups (based on the treatment outcomes and its baseline clinical correlates) that could be particularly benefitted from treatments including dietary zinc and iron supplements, such as the results of Bilici and colleagues [73]: the association found in this study between the low baseline zinc values with lower improvements in the ADHD measures suggested that zinc may be particularly important for the treatment of this disorder, improving and compensating borderline zinc nutrition. The study of Arnold and colleagues, which reanalyzed data from a randomized clinical trial with *n* = 18 children by comparing d-amphetamine with Efamol, assessed the moderate role of zinc [94], categorized the participants into three groups (as zinc-adequate, zinc-borderline, and zinc-deficient), and observed a linear relationship of d-amphetamine with zinc nutrition among the d-amphetamine group, and a quadratic (U-shaped) relationship of Efamol response to zinc (the benefit among this group was only found among the zinc-borderline condition). Finally, the research based on a secondary analysis of data collected in the study of Arnold and colleagues [71] analyzed the role of the status of iron, and determined that it was associated with ADHD symptom severity at baseline and with responses to psycho-stimulant treatment [95]. This study found an inverse correlation of serum ferritin with baseline inattention, hyperactivity, and impulsivity levels (as ferritin concentration increased, ADHD severity decreased), and also a significant correlation between serum ferritin and sensitivity to the amphetamine used for ADHD treatment (participants with higher ferritin concentrations required lower weight-adjusted doses of psychotropic treatment). These results were obtained after adjusting for the participants’ sex and age. The authors concluded that iron supplementation could contribute as a potential intervention to optimize responses to psycho-stimulants in children with low iron stores at baseline. Other studies have also showed that ferritin concentration is correlated with the dose of amphetamine necessary to reach an optimal clinical response [96]. However, results in this area are few, and future research should address the question of which are the optimal doses, as well as the benefits of using supplements of multiple vitamins and minerals.

## 5. Conclusions

Based on the results obtained in this systematic review, the specific role of dietary nutrients with zinc and iron still seems controversial for the treatment of ADHD, being most consistent with the evidence for zinc. Moreover, although the reviewed studies found a relationship between the use of dietary supplements containing these elements with the improvement of ADHD symptoms, neither the mono-causal role of a concrete specific nutritional deficiency among ADHD children nor the role of a concrete dietary nutrient in the management of this disorder were proven (as was reported quite recently [56]). Future controlled clinical trials are needed, examining the efficacy of mineral supplementation.

The results obtained in this systematic review should be considered for combining these multi-facet interventions with additional precise interventions, specifically focused on the nutritional state and clinical profile of each patient. The ultimate objective of these treatments, including medical, psychological, and diet plans, should be to decrease the levels of ADHD, but also its multiple negative correlates, in order to achieve better performance in vigilance and sustained attention tasks, executive functioning tasks (like planning and organization), set shifting and response monitoring, and complex problem-solving.

Finally, this study should be interpreted in consideration of certain limitations. First, a meta-analysis was not conducted. Systematic reviews of treatment plans often include statistical methods to summarize the results of independent research by combining the information of selected studies (through meta-analysis) with the aim to provide more precise estimates of the effects of health care. However, it must be outlined that not all topics allow a meta-analysis to be conducted, and these concrete systematic reviews use alternative methods of synthesis described as “narrative synthesis”. This was the case of our study: we did not employ meta-analyses because of the clinical heterogeneity of individual studies, with different intervention characteristics (for example, the iron/zinc supplements were administered in two trials within multi-supplements also including other trace elements, with different doses and/or frequency of doses, and with different durations), diverse outcome/effects measurement, different research settings, and differences in participants’ features (such as age, baseline disease severity, other comorbid conditions, or sociodemographic variables). Since current studies have proven that this heterogeneity could cause significant inaccurate summary effects and associated conclusions, and thus misleading decision-markers [97], we opted to conduct a narrative synthesis in accordance with the current guidelines. Other characteristics of the individual studies selected for this review also impact on the generalizability capacity of our review: the limited samples sizes, the convenience samples (non-probability sampling methods were used), and the frequency of men compared to women. 

## Figures and Tables

**Figure 1 nutrients-13-04059-f001:**
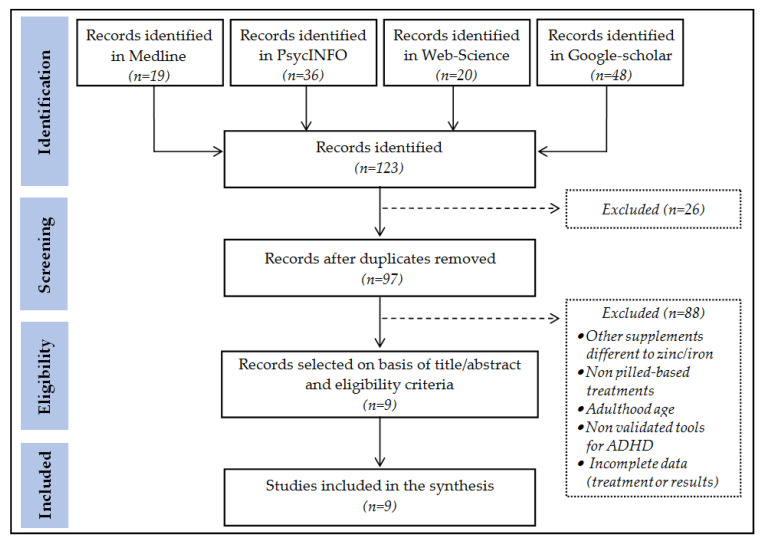
Search flow-chart.

**Table 1 nutrients-13-04059-t001:** Randomized controlled trials of supplements with iron and zinc for ADHD.

	Study	Refer.	*N*	Supp.	Dose	Age (Years)	Sex	Duration	Measures for ADHD	Results
1.	Noorazar et al. (2020)	[69]	60	Zinc	10 mg zinc/day	Range = 7–12 Mean = 9.67	20% Women 80% Men	6 weeks	● Conners Parent’s Questionnaire.	Zinc related with improvement, but only in the inattention factor
2.	Zamora et al. (2011)	[70]	40	Zinc	10 mg zinc/day	Range = 7–14 Mean = 9.8	22.5% Women 77.5% Men	6 weeks	● Conners Rating Scales-Revised.	Improvement in the Conner’s score, but only for the teacher version
3.	Arnold et al. (2011)	[71]	52	Zinc	15 mg zinc/day	Range = 6–14 Mean = 9.8	17.3% Women 82.7% Men	13 weeks	● Childen’s Interview for Psychiatric Syndromes (parent version). ● Parent and teacher behavioral ratings. ● Clinical Global Impressions (CGI). ● Conners’ Parent Rating Scale ● Neuropsychological cognitive-motor test battery.	Equivocal results for most measures. Only neuropsychological measures mostly favored zinc
4.	Akhondzadeh et al. (2004)	[72]	44	Zinc	15 mg zinc/day	Range = 5–11 Mean = 7.9	40.9% Women 59.1% Men	6 weeks	● ADH Rating Scale (ADHD-RS)	Parent and Teacher rating scale scores improved with zinc
5.	Bilici et al. (2004)	[73]	400	Zinc	40 mg zinc/day	Range = 6–14 Mean = 9.6	18% Women 82% Men	12 weeks	● Attention Deficit Hyperactivity Disorder Scale (ADHDS) ● Conners Teacher Questionnaire ● DuPaul Parent Ratings of ADHD	Zinc related with improvements in hyperactive, impulsive and socialization symptoms. No impact of zinc was observed for the attention deficit levels. A moderator effect with age and BMI was observed
6.	Konofal et al. (2008)	[74]	22	Iron	80 mg/day (ferrous sulfate, Tardyferon)	Range = 5–8 Mean = 5.9	21.7% Women 78.3% Men	12 weeks	● Conners’ Parent Rating Scale ● Conners’ Teacher Rating Scale ● Attention Deficit Hyperactivity Disorder Rating Scale (ADHD RS). ● Clinical Global Impression-Severity (CGI-S).	Iron related to improvement on the ADHD RS scale and the CGI-S score. Iron did not achieved improvements on the Conner’s tests
7.	Panahandeh et al. (2017)	[75]	42	Iron	5 mg/kg/day (ferrous sulfate)	Range = 5–15 Mean = 8.9	9.5% Women 90.5% Men	8 weeks	● Child Symptom Inventory-4 (CSI-4).	Iron related with higher decreases on the CSI-4 total and factor scores
8.	Rucklidge et al. (2018)	[76]	93	Zinc- Iron	Zinc (3.2 mg/capsule) Iron (0.9 mg/capsule). Dose: starting 3 and increasing to 12 capsules/day	Range = 5–15 Mean = 9.7	23.7% Women 76.3% Men	10 weeks	● Conners’ Parent Rating Scale (CPRS-R) ● Strengths and Difficulties Questionnaire (SDQ, Parents version) ● Strengths and Difficulties Questionnaire (SDQ, Teachers version)	Supplements related with improvements in inattentive levels. No contribution were observed on hyperactive-impulsive symptoms.
9.	Rucklidge et al. (2021)	[58]	38	Zinc- Iron	Zinc (3.2 mg/capsule) Iron (0.9 mg/capsule). Dose: starting 3 and increasing to 12 capsules/day	Range = 7–13 Mean = 10.1	21% Women 79% Men	10 weeks	● ADH Rating Scale IV (ADHD-RS-IV). ● Children’s Depression Rating Scale (CDRS) ● Children’s Globabl Assessment Scale (CGAS)	Differences in the ferritin levels achieved an interaction role for improving the ADHD severity levels.

Note: *N* = sample size. Supplem: supplement administered in the experimental Group. Refer: cite of the study. Dose: dosage for the supplement (zinc and iron). Results: related with the supplement (zinc and iron).

**Table 2 nutrients-13-04059-t002:** Assessment of the studies’ quality based on the CONSORT checklist.

			Title: Includes Design Type	Abstract: Structured-Complete	Introduction: Background	Introduction: Objectives-Hypothesis	Methods: Design Described	Methods: Participants	Methods: Interventions	Methods: Outcomes	Methods: Sample-Size Calculation—Power	Methods: Randomization	Methods: Implementation	Methods: Statistical Procedure	Results: participants Flow	Results: Numbers Analyzed	Results: Outcomes-Estimates	Discussion: Limitations	Discussion: Generalization	Discussion: Interpretation	Other Registration-Protocol-Funding
1.	Noorazar et al. (2020)	[69]	(+)	(+)	(+)	(P)	(+)	(+)	(+)	(+)		(+)	(+)	(+)	(+)	(+)	(P)	(+)	(P)	(+)	(+)
2.	Zamora et al. (2011)	[70]	(+)	(+)	(+)	(P)	(+)	(+)	(+)	(+)		(+)	(+)	(+)	(+)	(+)	(P)	(?)	(P)	(+)	(+)
3.	Arnold et al. (2011)	[71]	(+)	(+)	(+)	(+)	(+)	(+)	(+)	(+)		(+)	(+)	(+)	(+)	(+)	(+)	(+)	(P)	(+)	(+)
4.	Akhondzadeh et al. (2004)	[72]	(+)	(+)	(+)	(P)	(+)	(+)	(+)	(+)		(+)	(+)	(+)	(+)	(+)	(+)	(+)	(P)	(+)	(+)
5.	Bilici et al. (2004)	[73]	(+)	(+)	(+)	(+)	(+)	(+)	(+)	(+)		(+)	(+)	(+)	(+)	(P)	(P)	(?)	(P)	(+)	(+)
6.	Konofal et al. (2008)	[74]	(?)	(+)	(+)	(+)	(+)	(+)	(+)	(+)		(+)	(+)	(+)	(+)	(+)	(+)	(+)	(P)	(+)	(+)
7.	Panahandeh et al. (2017)	[75]	(?)	(+)	(+)	(P)	(+)	(+)	(+)	(+)		(+)	(+)	(+)	(+)	(+)	(+)	(+)	(P)	(+)	(+)
8.	Rucklidge et al. (2018)	[76]	(+)	(+)	(+)	(P)	(+)	(+)	(+)	(+)		(+)	(+)	(+)	(+)	(+)	(+)	(+)	(P)	(+)	(+)
9.	Rucklidge et al. (2021)	[58]	(?)	(+)	(+)	(P)	(+)	(+)	(+)	(+)		(+)	(+)	(+)	(+)	(+)	(+)	(+)	(P)	(+)	(+)

Note. (+) Green color cell: presented-reported. (P) Grey color cell: partially presented or reported with some limitations. (?) White color cell: not present or not reported.

**Table 3 nutrients-13-04059-t003:** Assessment of the studies’ quality based on the NICE checklist.

			A1. Adequate Randomization Method	A2. Adequate Concealment of Allocation	A3. Groups Comparable at Baseline	B1. Comparison Groups Received Same Care	B2. Participants Receiving Care Blind to Treatment	B3. Individuals Administering Care Blind to Allocation	C1. All Groups Followed Up for Equal Length of Time	C2a. How Many Participants Did Not Complete Treatment	C2b. Groups Comparable for Treatment Completion	C3. Participants in Each Were No Outcome Data Available	C3b. Groups Comparable Respect Availability of Outcome Data	D1. Adequate Length of Follow-Up	D2. Precise Definition of Outcome	D3. Reliable Method Used to Determine the Outcome	D4. Investigators Kept Blind to Participants Exposure	D5. Investigators Were Kept Blind to Confounding-Predictors
1.	Noorazar et al. (2020)	[69]	(+)	(+)	(+)	(+)	(+)	(+)	(+)	(+)	(+)	(+)	(+)	(+)	(+)	(+)	(?)	(?)
2.	Zamora et al. (2011)	[70]	(+)	(+)	(+)	(+)	(+)	(+)	(+)	(+)	(+)	(+)	(+)	(+)	(+)	(+)	(?)	(?)
3.	Arnold et al. (2011)	[71]	(+)	(+)	(+)	(+)	(+)	(+)	(+)	(+)	(+)	(+)	(+)	(+)	(+)	(+)	(?)	(?)
4.	Akhondzadeh et al. (2004)	[72]	(+)	(+)	(+)	(+)	(+)	(+)	(+)	(+)	(+)	(+)	(+)	(+)	(+)	(+)	(?)	(?)
5.	Bilici et al. (2004)	[73]	(+)	(+)	(?)	(+)	(+)	(+)	(+)	(+)	(+)	(+)	(+)	(+)	(+)	(+)	(?)	(?)
6.	Konofal et al. (2008)	[74]	(+)	(+)	(+)	(+)	(+)	(+)	(+)	(+)	(+)	(+)	(+)	(+)	(+)	(+)	(?)	(?)
7.	Panahandeh et al. (2017)	[75]	(+)	(+)	(+)	(+)	(+)	(+)	(+)	(+)	(+)	(+)	(+)	(+)	(+)	(+)	(?)	(?)
8.	Rucklidge et al. (2018)	[76]	(+)	(+)	(+)	(+)	(+)	(+)	(+)	(+)	(+)	(+)	(+)	(+)	(+)	(+)	(?)	(?)
9.	Rucklidge et al. (2021)	[58]	(+)	(+)	(+)	(+)	(+)	(+)	(+)	(+)	(+)	(+)	(+)	(+)	(+)	(+)	(?)	(?)

Note. (+) Green color cell: presented-reported. (?) White color cell: not present or not reported. Items A1 to A3: selection bias (systematic differences between the comparison groups). Items B1 to B3: performance bias (systematic differences between groups in the care provided, apart from the intervention under investigation). Items C1 to C3b: Attrition bias (systematic differences between the comparison groups with respect to loss of participants). Items D1 to D5: Detection bias (bias in how outcomes are ascertained, diagnosed or verified).

**Table 4 nutrients-13-04059-t004:** Assessment of the studies’ quality based on the CASP checklist.

			A1.Clearly Focused Question	A2. Use of Randomization Method	A3. Participants Accounted for	B4. Use of “Blinded” Methods	B5. Groups Similar at the Start of Randomization	B6. Each Study Group Received the Same Care	C7. Effects of Intervention Adequately Reported	C8. Precision Estimates Reported (CI or Other Effect Sizes)	C9. Cost-Effectiveness Analysis Was Done	D10. Applicability of the Results	D11. Intervention Provides Value
1.	Noorazar et al. (2020)	[69]	(+)	(+)	(+)	(+)	(+)	(+)	(+)	(?)	(?)	(+)	(+)
2.	Zamora et al. (2011)	[70]	(+)	(+)	(+)	(+)	(+)	(+)	(+)	(?)	(?)	(+)	(+)
3.	Arnold et al. (2011)	[71]	(+)	(+)	(+)	(+)	(+)	(+)	(+)	(+)	(?)	(+)	(+)
4.	Akhondzadeh et al. (2004)	[72]	(+)	(+)	(+)	(+)	(+)	(+)	(+)	(?)	(?)	(+)	(+)
5.	Bilici et al. (2004)	[73]	(+)	(+)	(+)	(+)	(+)	(?)	(+)	(?)	(?)	(+)	(+)
6.	Konofal et al. (2008)	[74]	(+)	(+)	(+)	(+)	(+)	(+)	(+)	(+)	(?)	(+)	(+)
7.	Panahandeh et al. (2017)	[75]	(+)	(+)	(+)	(+)	(+)	(+)	(+)	(?)	(?)	(+)	(+)
8.	Rucklidge et al. (2018)	[76]	(+)	(+)	(+)	(+)	(+)	(+)	(+)	(+)	(?)	(+)	(+)
9.	Rucklidge et al. (2021)	[58]	(+)	(+)	(+)	(+)	(+)	(+)	(+)	(+)	(?)	(+)	(+)

Note. (+) Green color cell: present-reported; (?) White color cell: not present or not reported. Items A1 to A3: the study design is valid for a randomized controlled trial. Items B4 to B6: the study methodology is sound. Items C7 to C9: the results are adequately reported. Items D10 to D11: the interpretation of the results is helpful.

**Table 5 nutrients-13-04059-t005:** Assessment of the risk of bias based on the Cochrane Risk of Bias 2 checklist.

			Randomization	Deviations from the Intended Intervention/s	Missing Outcome Data	Outcome Measurements	Selective Reporting	Incomplete Reporting	Study Power Calculation/Sample Size Justification
1.	Noorazar et al. (2020)	[69]	(?)	(+)	(+)	(+)	(+)	(+)	(−)
2.	Zamora et al. (2011)	[70]	(?)	(+)	(+)	(+)	(+)	(+)	(−)
3.	Arnold et al. (2011)	[71]	(?)	(+)	(+)	(+)	(+)	(+)	(−)
4.	Akhondzadeh et al. (2004)	[72]	(?)	(+)	(+)	(+)	(+)	(+)	(−)
5.	Bilici et al. (2004)	[73]	(?)	(+)	(+)	(+)	(+)	(+)	(−)
6.	Konofal et al. (2008)	[74]	(?)	(+)	(+)	(+)	(+)	(+)	(−)
7.	Panahandeh et al. (2017)	[75]	(?)	(+)	(+)	(+)	(+)	(+)	(−)
8.	Rucklidge et al. (2018)	[76]	(?)	(+)	(+)	(+)	(+)	(+)	(−)
9.	Rucklidge et al. (2021)	[58]	(?)	(+)	(+)	(+)	(+)	(+)	(−)

Note. (+) Green color cell: low risk of bias. (?) Grey color cell: moderate risk of bias (unclear reporting or partially reported). (−) Brown color cell: high risk of bias (non-reported). Randomization: low bias judged if both a method of randomization and concealing allocation was clearly described, along with no clear obvious baseline differences between groups. Deviation: low bias judged if study participants did not change between groups. Missing outcome: low bias judged if less than 20% of participants were lost between the beginning and the end of the treatment. Outcome measurement: low bias judged if validated tools were used for the main results (e.g., standardized questionnaires). Selective reporting: low bias judged if outcomes were pre-specified. Incomplete reporting: low bias judged if measurement methods, procedure, analysis and outcomes were specified in advance to the results section. Power calculation/sample size justification: low bias judged if this was performed in advance and attained in the study.

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
