# Peer review of "The Role of Iron and Zinc in the Treatment of ADHD among Children and Adolescents: A Systematic Review of Randomized Clinical Trials"

_nutrients, 2021, doi:10.3390/nu13114059_

Round 1
Reviewer 1 Report
This is a qualitative systematic review of RCTs for iron and/or zinc supplementation in children with ADHD.
The strongest part of the manuscript are the tables that lay out the guidelines CONSORT, NICE, CASP.
“Assessing the study quality (risk of bias)” was mentioned, but not included in relation to each of the studies.
How were the PRISMA and AMSTAR guidelines included in this manuscript?
- State which studies just used zinc or iron alone compared to zinc or iron with medication versus zinc or iron in combination with many other nutrients (Rucklidge);
- Say the amount of zinc or iron supplemented in each study (it’s done in some, but not in others – particularly for the multinutrient studies – report the total zinc or iron dosage range depending on how many capsules taken – could be added to Table 1 or in each section
Discussion: A sentence or two clearly stating the findings as they relate to the RCTs. What are the overall findings regarding supplementation – does it work?
Line 393, zinc is …..a dopamine reuptake inhibitor (not transporter); zinc and iron alter dopamine transporter functioning
This is a key point worth repeating in the Discussion section. In addition, nutrients act in a synergistic way, and therefore changes in one can have cascading effects in others, and therefore it
A “Limitations” section is needed in the Discussion section.
Conclusion: Starting at Line 430, the authors introduce new results. These points should be made earlier and summarized in the Discussion rather than new ideas presented for the first time
Lines 417- 419
Moreover, since no evidence supports the mono-causal role of a concrete specific nutritional deficiency among ADHD children neither the role of a concrete dietary nutrient in the management of this disorder, multimodal treatments are recommended including pharmacotherapy, psychotherapy and 420 psychoeducational plans.
It’s not accurate to say “no evidence” for single nutrient supplementation; there is evidence among those who are low. The last part (bolded) of this sentence is not reviewed in this manuscript, so it’s not appropriate to include this as “recommended.” In fact, Lines 124-125 say these treatments are not included. “Specifically excluded from this systematic review were non-pill-based treatment 124 modalities such as behavioral interventions, neurofeedback, restriction or alternative 125 food exclusion diets or chiropractic interventions”
Examples of some typos – there are many more
“evidence” is a single noun never plural (no “s”)
Line 52: Etiological researches outline – depending on intended meaning, should be “research outlines” or “researchers outline”
Line 129 “in” should be “on”
Line 197 omit “with”
Line 221 carrying out a cost-effectiveness 221 analysis cost).
Line 222, extra “)” at the end.
Table 3:
C3 and C3b on the top line have extra spaces – meaning is not clear
D3 and D5 on the top line have extra spaces; remove “were” in Ds
Table 4
C9 and D11 typos - word should be “cost effectiveness” or take out extra space
Line 243: omit “Any case”
Line 263 separate “this” and “trial”
Line 279 – should 6 years really say “6 weeks?”
Thank you for undertaking this work. In the manuscript's current form, it requires substantive edits for clarity.
Author Response
Response: Thank you very much for your helpful feedback on our manuscript. We have attempted to address your comments and have documented these changes below. In the revised manuscript, the changes were highlighted in yellow color and included using the Track-Changes-function.
This is a qualitative systematic review of RCTs for iron and/or zinc supplementation in children with ADHD. The strongest part of the manuscript are the tables that lay out the guidelines CONSORT, NICE, CASP. “Assessing the study quality (risk of bias)” was mentioned, but not included in relation to each of the studies.
Response: Thank you for highlighting this. We have now included a new sub-section in the results section with the assessment of the risk of bias. This sub-section includes also a new Table with the Cochrane Risk of Bias 2 checklist applied to the studies selected for the systematic review.
How were the PRISMA and AMSTAR guidelines included in this manuscript?
Response: We thank for this sound suggestion. We have now included a new Table with the results of the PRISMA checklist (Table S1, supplementary).
State which studies just used zinc or iron alone compared to zinc or iron with medication versus zinc or iron in combination with many other nutrients (Rucklidge);
Response: We appreciate the Reviewer’s constructive comment. We have now clarified the dose of zinc-iron in each study in Table 1, and also in the results section. We have also justified the inclusion of studies using multimineral-vitamin supplements which contained zinc and iron:
“Some randomized clinical studies have also valued the contribution of multimineral-vitamin supplements which include zinc and iron for the ADHD treatment in children. The rationale for selecting trails with multi-supplements therapy plans for ADHD was the low number of trails using simultaneously zinc and iron, and those identified used preparations with vitamins, dietary minerals and other nutritional elements. Therefore, considering these studies facilitates generalizability of the results related with the use of zinc-iron supplements”
Say the amount of zinc or iron supplemented in each study (it’s done in some, but not in others – particularly for the multinutrient studies – report the total zinc or iron dosage range depending on how many capsules taken – could be added to Table 1 or in each section
Response: We agree with this point made by the Reviewer here. We have now included the dosage in a new column in Table 1. In addition, we have clarified the dosage for the multinutrients studies in the results section:
“The study of Rucklidge and colleagues [78] in a sample of n=93 patients used a Daily Essential Nutrients (DEN, with Recommended Dietary Allowances, RDA), which contains a comprehensive range of micronutrients (13 vitamins, 17 minerals and 4 amono acids). The participants were instructed to titrate the dose up over a week, starting with 3 capsules per day and increasing up to 12 capsules per day (see Table 1 for the zinc-iron content)”
...
“Finally, the study of Rucklidge et al. published later and carried out on a sample of n=38 children used also DEN, with an initial dose of 3 capsules per day further increased to 12 capsules per day (see Table 1 for the zinc-iron content)”.
Discussion: A sentence or two clearly stating the findings as they relate to the RCTs. What are the overall findings regarding supplementation – does it work?
Response: We appreciate the Reviewer’s thoughtful comment. We have now rewritten the first paragraph of the discussion. It says now:
“This study constitutes a qualitative systematic review of the empirical evidence obtained in randomized clinical trials published since 2000 on the efficacy of zinc and iron supplementation among ADHD children and adolescents. Nine studies met the eligibility criteria and were selected. Results indicated that low zinc and iron levels were related with both higher baseline levels of ADHD severity and poorer treatment outcomes. Compared with the controlled-placebo conditions, the dietary supplements with zinc and iron were associated with improvements in the ADHD severity at the end of the treatments, although the effect size of the outcomes tended to be low and/or focused on specific ADHD symptoms/measures”.
Line 393, zinc is …..a dopamine reuptake inhibitor (not transporter); zinc and iron alter dopamine transporter functioning. This is a key point worth repeating in the Discussion section. In addition, nutrients act in a synergistic way, and therefore changes in one can have cascading effects in others, and therefore it
Response: Thank you for highlighting this. We have now reviewed the discussion section to clarify this topic.
A “Limitations” section is needed in the Discussion section.
Response: Thank you for highlighting this. We have now included a new paragraph at the end of the Discussion section, with the main limitations of the study. This new paragraph says:
“Finally, this study should be interpreted in consideration of certain limitations. First, a meta-analysis was not conducted. Systematic reviews of treatment plans often include statistical methods to summarize the results of the independent researches, by combining information of the selected studies (through meta-analysis) with the aim to provide more precise estimates of the effects of health care. However, it must be outlined that not all topics allow a meta-analysis to be conducted, and these concrete systematic reviews use alternative methods of synthesis described as “narrative synthesis”. This was the case of our study: we did not employ meta-analysis because of the clinical heterogeneity of individual studies, with different intervention characteristics (for example, the iron/zinc supplements were administered in two trails within multi-supplements including also other trace elements, with different dose and/or frequency of dose, and with different durations), diverse outcome/effects measurement, different research settings and differences in participants features (such as age, baseline disease severity, other comorbid conditions, or sociodemographic variables). Since current studies have proved that this heterogeneity could cause significant inaccurate summary effects and associated conclusions, misleading decision markers [97], we opted for conducted a narrative synthesis according with current guidelines. Other characteristics of the individual studies selected for this review also impact on the generalizability capacity of our review: the limited samples sizes, the convenience samples (non-probability sampling methods were used) and the frequency of men compared to women”.
Reference:
[97] Campbell M, McKenzie JE, Sowden A, et al. Synthesis without meta-analysis (SWiM) in systematic reviews: reporting guideline. BMJ. 2020;368:l6890. Published 2020 Jan 16. doi:10.1136/bmj.l6890
Conclusion: Starting at Line 430, the authors introduce new results. These points should be made earlier and summarized in the Discussion rather than new ideas presented for the first time
Response: Thank you for highlighting this. Based also in the comment made by other reviewer regarding the text in the conclusion, we have moved the last paragraph to the discussion.
Lines 417- 419
Moreover, since no evidence supports the mono-causal role of a concrete specific nutritional deficiency among ADHD children neither the role of a concrete dietary nutrient in the management of this disorder, multimodal treatments are recommended including pharmacotherapy, psychotherapy and 420 psychoeducational plans.
It’s not accurate to say “no evidence” for single nutrient supplementation; there is evidence among those who are low. The last part (bolded) of this sentence is not reviewed in this manuscript, so it’s not appropriate to include this as “recommended.” In fact, Lines 124-125 say these treatments are not included. “Specifically excluded from this systematic review were non-pill-based treatment 124 modalities such as behavioral interventions, neurofeedback, restriction or alternative 125 food exclusion diets or chiropractic interventions”
Response: Thank you for highlighting this. We have reviewed this paragraph in the conclusions section, and now it says:
“Based on the results obtained in this systematic review, the specific role of dietary nutrients with zinc and iron seems still controversial for the treatment of ADHD, being most consistent the evidence for zinc. Moreover, although the reviewed studies found a relationship between the use of dietary supplements containing these elements with the improvement of ADHD symptoms, the mono-causal role of a concrete specific nutritional deficiency among ADHD children neither the role of a concrete dietary nutrient in the management of this disorder were proved (as was reported quite recently [56]). Future controlled clinical trials are needed, examining the efficacy of mineral supplementation”.
Examples of some typos – there are many more
Response: We appreciate this helpful suggestion. The reviewed version of the manuscript has been now revised by a professor of English Language, who is a professional translator/corrector. All the typos indicated by the Reviewer have been corrected.
“evidence” is a single noun never plural (no “s”)
Line 52: Etiological researches outline – depending on intended meaning, should be “research outlines” or “researchers outline”
Line 129 “in” should be “on”
Line 197 omit “with”
Line 221 carrying out a cost-effectiveness 221 analysis cost).
Line 222, extra “)” at the end.
Table 3: a)C3 and C3b on the top line have extra spaces – meaning is not clear; b)D3 and D5 on the top line have extra spaces; remove “were” in Ds
Table 4: a) C9 and D11 typos - word should be “cost effectiveness” or take out extra space
Line 243: omit “Any case”
Line 263 separate “this” and “trial”
Line 279 – should 6 years really say “6 weeks?”
Reviewer 2 Report
--line 96: zinc and iron "low letters"
-line 198: are in these multi-supplements also vitamins and other trace elements included, besides iron and zinc ? If this is the case it is quite tricky to make firm conclusions.
-table 1 should include the number of the references, not only the year of publication. Otherwise it is hard to trace. Also for other tables, but especially the table 1 needs to be reconstructed in a smoother wayto look more clear.
-line 223 has to be clicked to the next page, so also for lines 226 and 232.
-lines 256-257: methylphenidate could contribute to... Is this decrease a result of decreased bioavailability of zinc ?
-line 266: what is a "zinc-wasting metabolism" ?
-line 301: do you mean by "pre-post decreases" a decrease in element level before and after the treatment ? This is a odd expression.
-line 306: "ferrous" instead of "ferrus"
-line 321: [74]
-line 327: multimineral-vitamin supplements. This is a tricky inclusion of studies, because of the interactions of all elements and vitamins in bioavailability, synergistic, antagonistic and additional actions.
-lines 341, 346 and 354 mention the space between some words
-line 417: here could be added ", "as was reported quite recently [56]"
-line 430: which are the criteria used to state the existence of subgroups ?
The zinc level in the blood ?
-
Author Response
Response: Thank you very much for your helpful feedback on our manuscript. We have attempted to address your comments and have documented these changes below. In the revised manuscript, the changes were highlighted in yellow color and included using the Track-Changes-function.
- Line 96: zinc and iron "low letters"
Response: Thank you for pointing this out. We have made the changes in reviewed manuscript.
- Line 198: are in these multi-supplements also vitamins and other trace elements included, besides iron and zinc? If this is the case it is quite tricky to make firm conclusions.
Response: Thank you for this feedback. We have now included a last paragraph with the main limitations of the study, which indicates as a specific difficulty the clinical heterogeneity and its implications:
“Finally, this study should be interpreted in consideration of certain limitations. First, a meta-analysis was not conducted. Systematic reviews of treatment plans often include statistical methods to summarize the results of the independent researches, by combining information of the selected studies (through meta-analysis) with the aim to provide more precise estimates of the effects of health care. However, it must be outlined that not all topics allow a meta-analysis to be conducted, and these concrete systematic reviews use alternative methods of synthesis described as “narrative synthesis”. This was the case of our study: we did not employ meta-analysis because of the clinical heterogeneity of individual studies, with different intervention characteristics (for example, the iron/zinc supplements were administered in two trails within multi-supplements including also other trace elements, with different dose and/or frequency of dose, and with different durations), diverse outcome/effects measurement, different research settings and differences in participants features (such as age, baseline disease severity, other comorbid conditions, or sociodemographic variables). Since current studies have proved that this heterogeneity could cause significant inaccurate summary effects and associated conclusions, misleading decision markers [97], we opted for conducted a narrative synthesis according with current guidelines. Other characteristics of the individual studies selected for this review also impact on the generalizability capacity of our review: the limited samples sizes, the convenience samples (non-probability sampling methods were used) and the frequency of men compared to women”.
Reference:
[97] Campbell M, McKenzie JE, Sowden A, et al. Synthesis without meta-analysis (SWiM) in systematic reviews: reporting guideline. BMJ. 2020;368:l6890. Published 2020 Jan 16. doi:10.1136/bmj.l6890
- Table 1 should include the number of the references, not only the year of publication. Otherwise it is hard to trace. Also for other tables, but especially the table 1 needs to be reconstructed in a smoother way to look more clear.
Response: Thank you for bringing this consideration to our attention. We have now included the number of the references into Table 1, and also in the other Tables of the study.
- Line 223 has to be clicked to the next page, so also for lines 226 and 232.
Response: Thank you for bringing this consideration to our attention. We have reviewed the editing and format of the final version of the manuscript after including all the suggestions and comments made by the Reviewers.
- Lines 256-257: methylphenidate could contribute to... Is this decrease a result of decreased bioavailability of zinc?
Response: Thank you for bringing this consideration to our attention. We have now described the two intervention plans (placebo group versus treated group) in the study of Noorazar and colleagues, and now have clarified that no differences between the groups were found for the dose of methylphenidate:
“The study of Noorazar and colleagues used the next intervention [69]: a) a dose of 0.5-1 mg/kg/day methylphenidate plus placebo in the control group; and b) a dose of 0.5-1 mg/kg/day methylphenidate plus 10 mg zinc (10 cc zinc sulfate syrup). No differences between the groups were observed regarding the dose of methylphenidate”.
- Line 266: what is a "zinc-wasting metabolism"?
Response: We thank for this sound comment. We have now clarified this:
“… children with ADHD may have zinc-wasting metabolism (defined as low levels of zinc related with potential deficits in the absorption or losses in the urine)”.
- Line 301: do you mean by "pre-post decreases" a decrease in element level before and after the treatment? This is a odd expression.
Response: Thank you for this constructive remark. We have now modified this sentence:
“Low pre-treatment zinc and free fatty acids values also were associated with fewer decreases in the ADHD symptom levels before and after the treatments”.
- Line 306: "ferrous" instead of "ferrus"
Response: Thank you for pointing this out. This error has been corrected.
- Line 321: [74]
Response: We have corrected the use of the square brackets in this sentence.
- Line 327: multimineral-vitamin supplements. This is a tricky inclusion of studies, because of the interactions of all elements and vitamins in bioavailability, synergistic, antagonistic and additional actions.
Response: We appreciate the Reviewer’s constructive comment. We have now justified the inclusion of studies using multimineral-vitamin supplements which contained zinc and iron:
“Some randomized clinical studies have also valued the contribution of multimineral-vitamin supplements which include zinc and iron for the ADHD treatment in children. The rationale for selecting trails with multi-supplements therapy plans for ADHD was the low number of trails using simultaneously zinc and iron, and those identified used preparations with vitamins, dietary minerals and other nutritional elements. Therefore, considering these studies facilitates generalizability of the results related with the use of zinc-iron supplements”
- Lines 341, 346 and 354 mention the space between some words.
Response: Thank you for pointing this out. These errors have been corrected.
- Line 417: here could be added ", "as was reported quite recently [56]"
Response: Thank you for pointing this out. We have added this comment and cite.
- Line 430: which are the criteria used to state the existence of subgroups?
Response: Thank you for pointing this out. We have now clarified that these subgroups should be based on the treatment outcomes and its baseline clinical correlates.
- The zinc level in the blood?
Response: Thank you for highlighting this. We have now reviewed the manuscript and change by “zinc level in blood tests.
Reviewer 3 Report
The manuscript presents a systematic review of clinical studies evaluating the potential benefits of incorporating iron and/or zinc supplementation into the therapy regime for treating child and adolescent ADHD. This is an interesting area of research, as current medication regimes do not significantly improve symptoms in many patients, and can also cause adverse side effects. Therefore the development of novel therapeutic regimes is of utmost importance. The submitted systematic review is well-implemented, with thorough documentation, and additionally includes several study quality assessment methods. However, there are several areas where the review could be improved, in order to increase its readability. My specific comments can be found below:
- The manuscript should be reviewed for English grammar and spelling, preferably by a native speaker. There are currently several spelling mistakes, and a few sentences which are hard to understand.
- Ensure the use of past tense when describing the methods (e.g. in the abstract, the present tense is used).
- Table 1 – boys is used for gender, and then men – please change to boys
- Table 1 – please change gender to sex, as gender should not be used to describe biological sex
- Please include reasons for exclusion in Figure 1 (standard reporting format for PRISMA flow charts)
- More discussion is needed regarding the limitations of the studies included in the review e.g. small sample sizes, male bias, different dosages
- The conclusion is too long – the last paragraph should be moved to the discussion
- Table 1 is very useful, but to increase its usefulness, it would be good to add the dosages used and also a short report of the main results found
Author Response
The manuscript presents a systematic review of clinical studies evaluating the potential benefits of incorporating iron and/or zinc supplementation into the therapy regime for treating child and adolescent ADHD. This is an interesting area of research, as current medication regimes do not significantly improve symptoms in many patients, and can also cause adverse side effects. Therefore the development of novel therapeutic regimes is of utmost importance. The submitted systematic review is well-implemented, with thorough documentation, and additionally includes several study quality assessment methods. However, there are several areas where the review could be improved, in order to increase its readability. My specific comments can be found below:
Response: The authors thank the Reviewer for your thoughtful comments and suggestions on our manuscript. We have considered all of the comments and incorporated them into the revised manuscript. Changes to the original document are highlighted in yellow color and included using the Track-Changes-function. An itemized point-by-point response to the Reviewers’ comments is presented below.
- The manuscript should be reviewed for English grammar and spelling, preferably by a native speaker. There are currently several spelling mistakes, and a few sentences which are hard to understand.
Response: We appreciate this helpful suggestion. The reviewed version of the manuscript has been now revised by a professor of English Language, who is a professional translator/corrector.
- Ensure the use of past tense when describing the methods (e.g. in the abstract, the present tense is used).
Response: Thank you for highlighting this. We have reviewed methods in the abstract to ensure the use of past tense.
- Table 1 – boys is used for gender, and then men – please change to boys
Response: Thank you for highlighting this. We have made this change in the Table.
- Table 1 – please change gender to sex, as gender should not be used to describe biological sex
Response: Thank you for highlighting this. We have made this change in the Table.
- Please include reasons for exclusion in Figure 1 (standard reporting format for PRISMA flow charts)
Response: We appreciate the Reviewer’s thoughtful comment. We have now added the reasons for exclusion in the flow-chart.
- More discussion is needed regarding the limitations of the studies included in the review e.g. small sample sizes, male bias, different dosages
Response: Thank you for this feedback. We have now included a last paragraph with the main limitations of the study, which indicates as a specific difficulty the clinical heterogeneity and its implications:
“Finally, this study should be interpreted in consideration of certain limitations. First, a meta-analysis was not conducted. Systematic reviews of treatment plans often include statistical methods to summarize the results of the independent researches, by combining information of the selected studies (through meta-analysis) with the aim to provide more precise estimates of the effects of health care. However, it must be outlined that not all topics allow a meta-analysis to be conducted, and these concrete systematic reviews use alternative methods of synthesis described as “narrative synthesis”. This was the case of our study: we did not employ meta-analysis because of the clinical heterogeneity of individual studies, with different intervention characteristics (for example, the iron/zinc supplements were administered in two trails within multi-supplements including also other trace elements, with different dose and/or frequency of dose, and with different durations), diverse outcome/effects measurement, different research settings and differences in participants features (such as age, baseline disease severity, other comorbid conditions, or sociodemographic variables). Since current studies have proved that this heterogeneity could cause significant inaccurate summary effects and associated conclusions, misleading decision markers [97], we opted for conducted a narrative synthesis according with current guidelines. Other characteristics of the individual studies selected for this review also impact on the generalizability capacity of our review: the limited samples sizes, the convenience samples (non-probability sampling methods were used) and the frequency of men compared to women”.
Reference:
[97] Campbell M, McKenzie JE, Sowden A, et al. Synthesis without meta-analysis (SWiM) in systematic reviews: reporting guideline. BMJ. 2020;368:l6890. Published 2020 Jan 16. doi:10.1136/bmj.l6890
- The conclusion is too long – the last paragraph should be moved to the discussion
Response: We appreciate the Reviewer’s constructive comment. We have now made this modification.
- Table 1 is very useful, but to increase its usefulness, it would be good to add the dosages used and also a short report of the main results found
Response: We agree with this point made by the Reviewer here. We have now included the next new columns in Table 1: a) reference of each study; b) dose; and c) main treatment outcomes related with the use of the zinc-iron supplements.
Reviewer 4 Report
This study carried out a qualitative systematic review on the efficacy of zinc and iron among ADHD children and adolescence. The inclusion criteria of the relevant papers are clearly demonstrated. The contents of each paper are concisely summarized. This paper would provide clinicians with the handy current reference on the evidence of supplementation of two trace minerals. I want to know just one thing. Why didn’t authors employ a meta-analysis in this study?
Author Response
Response: The authors thank the Reviewer for the careful reading of the manuscript and the thoughtful comment regarding the use of a meta-analysis.
Changes to the original document are highlighted in yellow color and included using the Track-Changes-function. An itemized point-by-point response to the Reviewers’ comments is presented below.
We have now included the next content in the limitations section:
“Finally, this study should be interpreted in consideration of certain limitations. First, a meta-analysis was not conducted. Systematic reviews of treatment plans often include statistical methods to summarize the results of the independent researches, by combining information of the selected studies (through meta-analysis) with the aim to provide more precise estimates of the effects of health care. However, it must be outlined that not all topics allow a meta-analysis to be conducted, and these concrete systematic reviews use alternative methods of synthesis described as “narrative synthesis”. This was the case of our study: we did not employ meta-analysis because of the clinical heterogeneity of individual studies, with different intervention characteristics (for example, the iron/zinc supplements were administered in two trails within multi-supplements including also other trace elements, with different dose and/or frequency of dose, and with different durations), diverse outcome/effects measurement, different research settings and differences in participants features (such as age, baseline disease severity, other comorbid conditions, or sociodemographic variables). Since current studies have proved that this heterogeneity could cause significant inaccurate summary effects and associated conclusions, misleading decision markers [97], we opted for conducted a narrative synthesis according with current guidelines. Other characteristics of the individual studies selected for this review also impact on the generalizability capacity of our review: the limited samples sizes, the convenience samples (non-probability sampling methods were used) and the frequency of men compared to women”.
Reference:
[97] Campbell M, McKenzie JE, Sowden A, et al. Synthesis without meta-analysis (SWiM) in systematic reviews: reporting guideline. BMJ. 2020;368:l6890. Published 2020 Jan 16. doi:10.1136/bmj.l6890
Round 2
Reviewer 2 Report
-line 381: "amino" instead of "amono"
-line 403: "copper" instead of "cooper"
-line 821: mention the name of all authors, conform all other citations.